# Domain Adaptation using Silver Standard Masks for Lateral Ventricle Segmentation in FLAIR MRI

**Owen Crystal**[1]                                      OCRYSTAL@TORONTOMU.CA

**Pejman J. Maralani**[2]                          PEJMAN.MARALANI@UTORONTO.CA

**Sandra E. Black**[3]                            SANDRA.BLACK@SUNNYBROOK.CA

**Alan R. Moody**[2]                              ALAN.MOODY@SUNNYBROOK.CA

**April Khademi**[1,4,5]                             AKHADEMI@TORONTOMU.CA

[1] *Electrical, Computer and Biomedical Eng., Toronto Metropolitan University, Toronto, ON, Can.*

[2] *Department of Medical Imaging, University of Toronto, Toronto, ON, Can.*

[3] *Department of Neurology, University of Toronto, Toronto, ON, Can.*

[4] *Keenan Research Center, St. Michael's Hospital, Toronto, ON, Can.*

[5] *Institute of Biomedical Engineering, Science and Technology (iBEST), Toronto, ON, Can.*

**Editors:** Accepted for publication at MIDL 2023

## Abstract

Lateral ventricular volume (LVV) is an important biomarker for clinical investigation. We present the first transfer learning-based LVV segmentation method for fluid-attenuated inversion recovery (FLAIR) MRI. To mitigate covariate shifts between source and target domains, this work proposes an domain adaptation method that optimizes performance on three target datasets. Silver standard (SS) masks were generated from the target domain using a novel conventional image processing ventricular segmentation algorithm and used to supplement the gold standard (GS) data from the source domain, Canadian Atherosclerosis Imaging Network (CAIN). Four models were tested on held-out test sets from four datasets: 1) SS+GS: trained on target SS masks and fine-tuned on source GS masks, 2) GS+SS: trained on source GS masks and fine-tuned on target SS masks, 3) trained on source GS (GS CAIN Only) and 4) trained on target SS masks (SS Only). The SS+GS model had the best and most consistent performance (mean DSC = 0.89, CoV = 0.05) and showed significantly ($p < 0.05$) higher DSC compared to the GS-only model on three target domains. Results suggest pre-training with noisy labels from the target domain allows the model to adapt to the dataset-specific characteristics and provides robust parameter initialization while fine-tuning with GS masks allows the model to learn detailed features. This method has wide application to other medical imaging problems where labeled data is scarce, and can be used as a per-dataset calibration method to accelerate wide-scale adoption.

**Keywords:** Domain adaptation, transfer learning, ventricle segmentation, silver standards

## 1. Introduction

Enlarged ventricles have been associated with deficits in cognition and memory (Ertekin et al., 2016). Manual analysis of the lateral ventricular volume (LVV) is labourious and subjective and automated tools are the optimal alternative. Previous studies have primarily segmented the LVV using T1-weighted MRI images and achieved a mean Dice Similarity Coefficient (DSC) ranging from 0.84-0.88 using deep learning (Ertekin et al., 2016).

While T1-MRI is used for the structural analysis of anatomy, fluid-attenuated inversion recovery (FLAIR) MRI is more readily analyzed for white matter lesions, which are related to stroke, ischemia, and dementia (Wardlaw et al., 2015). FLAIR MRI is routinely used in clinical practice by neuroradiologists, radiologists, and neurologists, and therefore, biomarker tools for this sequence have high translation potential (Khademi et al., 2021). FLAIR may have the potential to better segment the LVV due to the nulled CSF signal increasing the CSF-to-tissue contrast (Soltanian-Zadeh and Peck, 2001) (Narayanan et al., 2020). A FLAIR-only LVV segmentation deep learning model achieved a mean DSC of 0.83 when testing on multi-centre data showing feasibility (Dubost et al., 2020). This work aims to develop and validate the first deep learning-based segmentation method for the LVV in mutli-centre FLAIR MRI using transfer learning.

Convolutional neural networks (CNN) excel at image segmentation due to their ability to extract high-dimensional and nonlinear features. The challenge, however, is when they are applied to images from a distribution outside the training dataset, there is a drop in performance. This covariate shift is common in medical imaging due to different scanning devices (hardware/software), acquisition parameters, and protocols (Ioffe and Szegedy, 2015b) (Huang et al., 2006) (Um et al., 2019). Generalization gaps create translational barriers since algorithms created by vendors cannot be widely deployed and adopted. Ideally, annotations from the target site could supplement training sets, but generating gold standard (GS) labeled ground truths for medical imaging is laborious and expensive.

To overcome these challenges, this work proposes a novel conventional image processing-based (IPB) LVV segmentation method that automatically generates silver standard (SS) masks in the unlabeled target domain to supplement GS data to improve target domain performance. As will be shown, the method can be used to calibrate AI tools on a per-site or per-dataset basis, in an easy manner that improves generalization and mitigates multi-centre variability (Khademi et al., 2020). The concept of using SS masks to increase the sample size of the training set has been successfully implemented in the application of intracranial volume (ICV) segmentation (Lucena et al., 2019). This work offers the following contributions: (1) We propose the first IPB LVV segmentation method. (2) We demonstrate how to leverage SS masks from an unlabeled target domain to improve performance and consistency in the target domain(s) for domain adaptation. (3) Propose the first ever FLAIR-only deep learning-based LVV segmentation model using transfer learning. (4) Propose a generic method for per-site calibration for any medical imaging AI problem.

## 2. Methodology
### 2.1. Data

Four multi-centre, multi-scanner datasets were used in this work. The first is from Alzheimer's Disease Neuroimaging Initiative (ADNI) (Jr et al., 2008), which is a public repository for studying dementia. The second dataset is from the Canadian Atherosclerosis Imaging Network (CAIN), which is a vascular disease study (Tardif et al., 2013). The third dataset is a dementia cohort from the Canadian Consortium on Neurodegeneration in Aging (CCNA)(Chertkow et al., 2019). The last dataset is from the Ontario Neurodegenerative Disease Research Initiative (ONDRI), which is also a dementia dataset (Ramirez et al., 2020). CAIN was used as the source domain and ADNI, CCNA, and ONDRI were used as the target domains. Datasets have varying acquisition parameters (Table 1), resulting in a

Table 1: FLAIR MRI ground truth data. All data is 3T and 3-5mm slice thickness.

| Patient Information | | | | | | |
|---|---|---|---|---|---|---|
| Database | Disease | Volumes | Images | Patients | Centres | LVV $\pm$ SD (mL) |
| ADNI | Dementia | 30 | 1260 | 30 | 12 | 64.67 $\pm$ 37.65 |
| CAIN | Vascular | 80 | 3840 | 80 | 8 | 30.22 $\pm$ 16.06 |
| CCNA | Dementia | 30 | 1440 | 30 | 9 | 42.26 $\pm$ 31.78 |
| ONDRI | Dementia | 30 | 1440 | 30 | 6 | 37.52 $\pm$ 13.31 |
| Total | All | 170 | 7980 | 170 | 33 | 39.71 $\pm$ 15.24 |
| Acquisition Parameters | | | | | | |
| Database | GE/Philips/Siemens | TR (ms) | TE (ms) | TI (ms) | X Spacing (mm) | Y Spacing (mm) |
| ADNI | 10/10/10 | 9000-11000 | 90-154 | 2250-2500 | 0.8594 | 0.8594 |
| CAIN | 20/20/20 | 9000-11000 | 117-150 | 2200-2800 | 0.4285-1 | 0.4285-1 |
| CCNA | 10/10/10 | 9000-9840 | 117-148 | 2250-2500 | 0.9375 | 0.9375 |
| ONDRI | 10/10/10 | 2250-100000 | 90-12810 | 2250-100000 | 0.92-0.94 | 0.89-0.94 |
| Total | 50/50/50 | 2250-100000 | 90-12810 | 2250-100000 | 0.4295-1.2 | 0.4295-1.2 |

diverse clinical sample. There are 30 GS volumes for each target domain (90 volumes, 4500 images) and 80 GS volumes (3840 images) from CAIN, the source domain.

## 2.2. Image Pre-Processing

All volumes underwent a series of standard image pre-processing techniques, including bias field correction to remove any low frequencies corrupting the MRI images and intensity standardization to align intensity distributions across the dataset (Reiche et al., 2019). ICV segmentation was completed using a MultiResUNET CNN was used to remove all non-brain tissues (i.e., skull, orbital cavities, and skin) (DiGregorio et al., 2021).

## 2.3. Gold Standard (GS) Annotations

The GS annotations were done by a medical student trained by a radiologist for ventricle annotations in ITKSnap. If the choroid plexus obscured the edge of the ventricle, the delineation included the choroid plexus within the ventricles. The septum pellucidum was excluded from the annotations. Areas below the anterior-posterior commissure line were eliminated to exclude the 3rd and 4th ventricles as well as the inferior horns of the lateral ventricle(Dwyer et al., 2017). To verify the annotations, a second medical student was trained with the same protocol and outlined the ventricles of 20 volumes. Inter-rater agreement was measured as mean DSC (0.93) and average volume difference (3%) indicating high repeatability and accuracy.

## 2.4. Target Domain Silver Standard (SS) Masks

A conventional image processing method was designed to generate SS LVV masks from the target domains. Intensity standardization aligns the intensity histograms across all datasets, which permits the same threshold to be applied to volumes from multi-centre data. A threshold of 200 was applied to intensity standardized images to remove gray and white matter and isolate the total CSF (ventricles + subarachnoid space) (Bahsoun et al., 2014). The thresholds and standardized methods have been validated on over 250,000 images from 100 centres (Reiche et al., 2019). A 25mm disk erosion kernel was applied to the brain tissue mask as per the work in (Zhong et al., 2014), which was then multiplied by the total CSF mask. Next, a largest object search was used to remove the subarachnoid CSF, thus isolating the ventricles. Hole filling was applied to the largest object. To ensure the detected objects are also located near the centre of the brain (with respect to the frontal plane), the distances between the centre of mass of the brain and each detected object

were calculated. If the distance was greater than 25mm (Zhong et al., 2014), this object was discarded and the next largest object was retained as the ventricles. Pixels below the anterior-posterior commissure line were removed (Dwyer et al., 2017). This method can be used to easily and efficiently generate SS training data from any unlabeled target imaging centre. The performance of the IPB method on ground truth annotations from the four multi-centre datasets (120 volumes) will be investigated to highlight performance.

## 2.5. Deep Learning-based Ventricular Segmentation

A 2D U-Net was used as the CNN architecture for this work as it is a fast and accurate deep learning technique that has achieved high performance on biomedical data (Ronneberger et al., 2015), (Hwang et al., 2019), (Thakur et al., 2020). The encoding path consists of convolutional and max pooling layers to perform feature extraction. The decoding path contains units of convolutional and transposed convolutional layers and skip connections to recapture spatial context (Long et al., 2015). U-Net contains five levels where the filter depth is doubled during each downsampling block (via max pooling) and halved during each upsampling block (via transposed convolution). In this work, U-Net was implemented with batch normalization layers succeeding convolutional layers (Ioffe and Szegedy, 2015a) to accelerate convergence and improve generalization via a modest regularization effect. Adam optimizer was used with a learning rate of 1e-4, batch size of 16 with 50 epochs, a 256x256 input resolution, and a generalized dice loss for all models and experiments. Data augmentations were applied for rotation, scaling, shearing, scaling, and translation. As per other FLAIR segmentation algorithms (DiGregorio et al., 2021) (Khademi et al., 2021), we chose the model from the 50th epoch. To mitigate overfitting, we used early stopping (validation loss does not improve after 15 epochs) and augmentation. Models were trained on a NVIDIA GeForce RTX 3090 Ti GPU with 32GB of RAM.

## 2.6. Transfer Learning

Transfer learning and fine-tuning neural networks is an effective method to enhance performance by transferring knowledge of a network trained on one dataset, onto another dataset or problem domain. Pre-trained networks from natural images may not be beneficial for medical image applications, due to differences in image properties and the covariate shift that is pervasive in medical imaging. In this work, we use the IPB method to generate SS masks in the unlabeled target domain, and use them to generate two models: 1) trained on target SS masks and fine-tuned on source GS masks (SS+GS), 2) trained on source GS masks and fine-tuned on target SS masks (GS+SS). The same model parameters from Section 2.5 are used for fine-tuning. The GS+SS is a more conventional experimental setup for transfer learning whereas we hypothesize that for the SS+GS model, using the SS to pre-train a base model will allow the model to learn low-level ventricular features and provide an optimal parameter initialization. The GS masks will be used to fine-tune the SS+GS model to capture more precise details. In Amiri et al. the authors determined that fine-tuning all of the layers of the U-Net architecture yielded the optimal segmentation performance and therefore will be adopted in this work (Amiri et al., 2020).

### 2.7. Performance Metrics

To measure segmentation performance and consistency of segmentations, the Dice Similarity Coefficient (DSC) and coefficient of variation (CoV) of the DSC were used. To investigate volume-based metrics, linear regression and correlation coefficients were used to compare predicted LVV vs. ground truth LVV, as well as Bland Altman plots. ANOVA and Tukey's posthoc on DSC were used to investigate if models are performing better on target data compared to GS CAIN Only.

### 2.8. Experimental Setup

Four types of models are examined for ventricular segmentation on target data. The first model is GS CAIN Only, which is trained with 50 CAIN (source) GS masks. The second model is trained using SS masks from the target dataset (SS Only). The last two models are the SS+GS and GS+SS models that use GS CAIN annotations along with the SS masks. Table 4 summarizes the data for each model. SS masks (50/100/150/200) were extracted using the method described in Section 2.4 for each target (ADNI, CCNA, and ONDRI) and source (CAIN) datasets. Volumes were stratified according to scanner vendors (Siemens, GE, Phillips). A total of 49 deep-learning models were trained. Thirty volumes for each target and source dataset (120 volumes in total) with GS annotations were held-out (non-overlapping, unique subjects) for testing performance.

## 3. Results

Qualitative results for all algorithms are shown in Fig. 3. There are false negatives from the GS CAIN Only model when testing on inferior slices (where LVs are more morphologically complex) or when LVs were relatively narrow, especially in target data. DSC and CoV for all models, including the IPB algorithm, on 30 held-out CAIN volumes (source domain) and all 90 volumes from target domains (CCNA, ADNI, ONDRI) are shown in Fig.1 and summarized in Table 2. The IPB algorithm (SS masks) achieved a mean DSC=0.82 and CoV=0.10 on all 120 held-out volumes.

### 3.1. Source Domain Performance

DSC and CoV for all models on 30 held-out CAIN volumes (source domain) show that the SS+GS model yielded the most accurate (DSC = 0.89) and consistent (CoV = 0.04) segmentations. The GS+SS model showed a decrease in performance compared to the GS CAIN Only model. While segmentation performance and consistency were slightly better in the SS+GS fine-tuned model, there was no significant difference ($p = 0.96$) when compared to GS CAIN Only. There was no difference in performance across the models with different numbers of SS masks in the GS+SS or SS+GS models.

### 3.2. Target Domain Performance

All four SS+GS models yielded the highest DSC and lowest CoV when testing on target datasets. ANOVA and Tukey's posthoc found significant (DSC) performance improvements when comparing the SS+GS models to GS CAIN Only models ($p < 0.05$). ANOVA also indicated no difference ($p = 0.99$) in DSC performance for different amounts of SS masks

Table 2: Performance on held-out testing volumes. Bold is best for that dataset and *
indicates model is significantly ($p < 0.05$) different from GS CAIN Only.

| Model (SS Masks) | CAIN | | ADNI | | CCNA | | ONDRI | |
|---|---|---|---|---|---|---|---|---|
| | DSC | CoV | DSC | CoV | DSC | CoV | DSC | CoV |
| IPB (N/A) | 0.83 | 0.09 | 0.82 | 0.11 | 0.81 | 0.14 | 0.84 | 0.08 |
| GS CAIN Only (0) | 0.88 | 0.05 | 0.85 | 0.08 | 0.85 | 0.17 | 0.85 | 0.15 |
| SS Only (50) | 0.80 | 0.15 | 0.81 | 0.11 | 0.81 | 0.13 | 0.84 | 0.07 |
| SS Only (100) | 0.78 | 0.15 | 0.80 | 0.15 | 0.83 | 0.10 | 0.84 | 0.08 |
| SS Only (150) | 0.85 | 0.09 | 0.84 | 0.10 | 0.83 | 0.10 | 0.84 | 0.08 |
| SS Only (200) | 0.84 | 0.11 | 0.83 | 0.10 | 0.82 | 0.11 | 0.84 | 0.08 |
| GS+SS (50) | 0.86 | 0.06 | 0.85 | 0.12 | 0.88* | 0.06 | 0.86 | 0.13 |
| GS+SS (100) | 0.88 | 0.06 | 0.86 | 0.08 | 0.89* | 0.08 | 0.86 | 0.11 |
| GS+SS (150) | 0.85 | 0.07 | 0.85 | 0.09 | 0.88 | 0.07 | 0.84 | 0.14 |
| GS+SS (200) | 0.85 | 0.06 | 0.80 | 0.17 | 0.88 | 0.07 | 0.86 | 0.15 |
| SS+GS (50) | **0.89** | **0.04** | 0.88* | **0.05** | 0.89* | 0.05 | 0.88* | **0.05** |
| SS+GS (100) | **0.89** | **0.04** | 0.88 | 0.06 | **0.90*** | **0.04** | **0.89*** | **0.05** |
| SS+GS (150) | **0.89** | 0.05 | 0.88 | 0.06 | **0.90*** | 0.06 | **0.89*** | **0.05** |
| SS+GS (200) | **0.89** | **0.04** | **0.89*** | **0.05** | 0.88* | 0.05 | 0.88* | **0.05** |

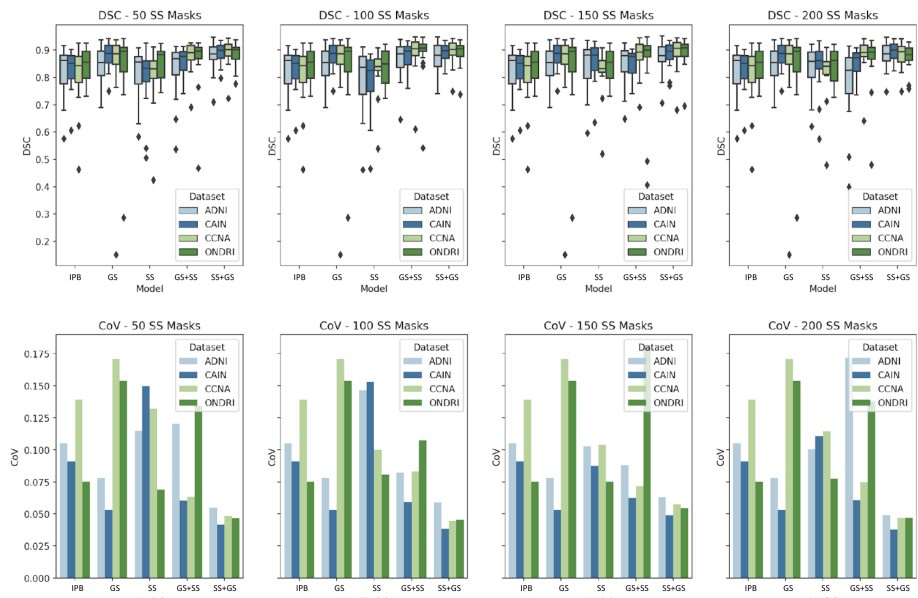

Figure 1: DSC and CoV for each model by number of SS masks.

in the SS+GS models. The GS+SS model showed significant differences in comparison to
the GS CAIN Only model when testing on CCNA data.

The SS Only, GS+SS, and GS+SS models with 100 SS masks were retained and compared further to the GS CAIN Only models. All four models were tested on the held-out
target datasets and linear regression and Bland-Altman analysis between the predicted and
ground truth LVV is shown in Fig. 2. Table 3 summarizes linear regression results on all

datasets and Table 5 shows the results by dataset. The SS+GS model yielded the highest correlation when testing on all the data, with the least amount of difference compared to the ground truths on Bland-Altman plots (lowest offset between the predicted LVV and true LVV as well as tighter bounds) indicating less variability and better consistency. Interestingly, the lowest intercept and highest slope were for the IPB approach which supports the use of this method for SS generation. The second best was the SS+GS model.

Table 3: Linear regression between predicted and true LVV on all data.

| Dataset | Model | $R^2$ | Intercept | Slope |
|---------|-------|-------|-----------|-------|
|          | IPB          | 0.77 | **3.97**  | **0.83** |
|          | GS CAIN Only | 0.80 | 11.72     | 0.64 |
| All Data | SS Only      | 0.68 | 11.92     | 0.48 |
|          | GS+SS        | 0.81 | 10.46     | 0.67 |
|          | SS+GS        | **0.84** | 9.93  | 0.72 |

## 4. Discussion

Although previous works primarily used T1-weighted MRI for LVV segmentation (Ertekin et al., 2016) (Wu and Tanga, 2021), results in this work show FLAIR is an effective imaging sequence for LVV segmentation. This can be attributed to the increased CSF-to-tissue contrast and strong performance of CNN-based segmentation schemes. In comparison to the one existing FLAIR-only technology, our SS+GS, GS+SS, and GS CAIN Only models outperforms on multi-centre, out-of-distribution data, with mean DSCs of 0.89, 0.86, and 0.85 respectively, compared to 0.83 (Dubost et al., 2020).

Previous technologies have demonstrated that transfer learning can be successfully used when the source and target data are from different domains (e.g., natural images to medical images) (Amiri et al., 2020) (Cheplygina, 2019) (Tajbakhsh et al., 2016). The results in this work show that pre-training with SS masks from the target domain and fine-tuning with source domain GS masks significantly improved overall performance and consistency in the target domain compared to source models. We hypothesize the model learns dataset- and problem-specific global trends from target SS masks that are injected into the model. As shown by SS Only models, the target data is not enough, and that is because the SS masks can have noise, which is cleaned up by the GS on fine-tuning. This claim is further supported when comparing the results of SS+GS and the GS+SS models. It appears that fine-tuning the GS-trained model with SS masks allows the model to learn the noisy aspects of the SS masks, thus "forgetting" the detailed features the GS masks possess. It was concluded that exposing the model to SS masks from the target domain in training helps to adapt to other domains and better improves performance in comparison to pre-training on source data and fine-tuning on target data. This also highlights that instead of using out-of-problem-domain pre-trained models like ImageNet, this domain adaptation technique poses a new way to generate pre-trained models that can be trained on SS data. While we show good performance for LVV applications, this could have wide implications for other pre-training medical imaging applications.

This work was validated on four large, multi-centre clinical datasets (CAIN, ADNI, CCNA, and ONDRI) which is a huge benefit of the work. The method was successfully applied to multiple target domains with similar results, demonstrating the robustness of

this domain-adaptation pipeline. The SS+GS model has the best performance on the target domain (with roughly the same performance) and a 4% (significant) increase in DSC compared to the GS model. This improvement can be seen visually in Fig. 3 where the GS CAIN Only models underestimate small ventricular areas, which is exacerbated in the out-of-domain targets. The drop in performance of the GS CAIN Only model when testing on volumes outside of the distribution is similar to the drop seen in (Dubost et al., 2020) as they achieved a mean DSC of 0.89 when testing on single-centre data, and a mean DSC of 0.83 when testing on multi-centre data. CAIN, largely a non-dementia cohort, acting as the source domain, could have led to performance degradation when testing on the target domains, which consisted of demented subjects. Future work will examine the impacts of using different cohorts as the source domain and the impact of disease. The SS+GS model improved segmentation performance in the target datasets, specifically on narrow LVVs and LVVs located in both inferior and superior slices, where the morphological shape becomes quite complex. This may be because the SS+GS model was exposed to more site-specific characteristics that helped the model learn data distribution properties (which improved predictions). The SS+GS model has the smallest deviation with a few outliers with the lowest mean volume difference, which indicates robustness, consistency, and reliability. Pre-training using SS masks provided a better parameter initialization, thus leading to a more generalized and accurate model. Results indicate the model benefits from being exposed to dataset-specific features and variables in training, albeit via noisy labels. These features/variables include many that could be related to covariate shifts (intensity distributions, imaging parameters, and/or image resolution). Future works could investigate optimization in the presence of noisy labels. The IPB algorithm showed strong intercept and slope values but relatively poor correlation coefficients (except when testing on ONDRI) suggesting that the algorithm is over-segmenting and under segmenting. However, the DSC is moderate (DSC = 0.82) indicating this is a good tool to use for SS generation. The IPB method provides a fast method of generating labels as it can annotate 100 SS masks in approximately 5-7 minutes, whereas manual annotation takes 30-60 minutes for a single imaging volume. A limitation of the IPB algorithm used to generate the SS masks is it is dependent on the ventricles being located near the middle of the brain. Although this is usually the case, a significant midline shift may occur due to a tumour or hematoma (Liao et al., 2018), which could lead to false negatives within the IPB algorithm. Therefore, SS masks should not be generated for images without pathology.

## 5. Conclusion

This work proposes the first FLAIR MRI-only lateral ventricular volume (LVV) segmentation method using transfer learning to pre-train a model on SS masks and fine-tune with GS masks. A novel domain adaptation technique for LVV segmentation in FLAIR MRI was designed to improve segmentation accuracy and mean volume difference across three multi-centre target datasets by using SS masks and transfer learning. The proposed domain adaptation methodology improved both the accuracy and consistency of the segmentation compared to predictions from the source domain model. The proposed method is an easy and effective method to design dataset-specific pre-trained models using the vast amounts of unlabeled target data available in imaging centres.

## Acknowledgments

We acknowledge the Natural Sciences and Engineering Research Council (NSERC) of Canada for funding this research through the Discovery Grant program.

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

## Appendix A. Supplemental Data

Table 4: Data splits of SS and GS masks for model training, tuning, and testing. GS training masks are from source domain (CAIN). SS Only and SS+GS models used the SS masks for training and the GS+SS model used them for fine-tuning.

| Model | Dataset | SS Masks | GS Tune | GS Train | Target Test |
|---|---|---|---|---|---|
| IPB | All | 0 | 0 | 0 | 120 |
| GS CAIN Only | All | 0 | 0 | 50 | 120 |
| SS Only | CAIN | 50/100/150/200 | 0 | 0 | 30 |
| SS Only | ADNI | 50/100/150/200 | 0 | 0 | 30 |
| SS Only | CCNA | 50/100/150/200 | 0 | 0 | 30 |
| SS Only | ONDRI | 50/100/150/200 | 0 | 0 | 30 |
| GS+SS | CAIN | 50/100/150/200 | 0 | 50 | 30 |
| GS+SS | ADNI | 50/100/150/200 | 0 | 50 | 30 |
| GS+SS | CCNA | 50/100/150/200 | 0 | 50 | 30 |
| GS+SS | ONDRI | 50/100/150/200 | 0 | 50 | 30 |
| SS+GS | CAIN | 50/100/150/200 | 50 | 0 | 30 |
| SS+GS | ADNI | 50/100/150/200 | 50 | 0 | 30 |
| SS+GS | CCNA | 50/100/150/200 | 50 | 0 | 30 |
| SS+GS | ONDRI | 50/100/150/200 | 50 | 0 | 30 |

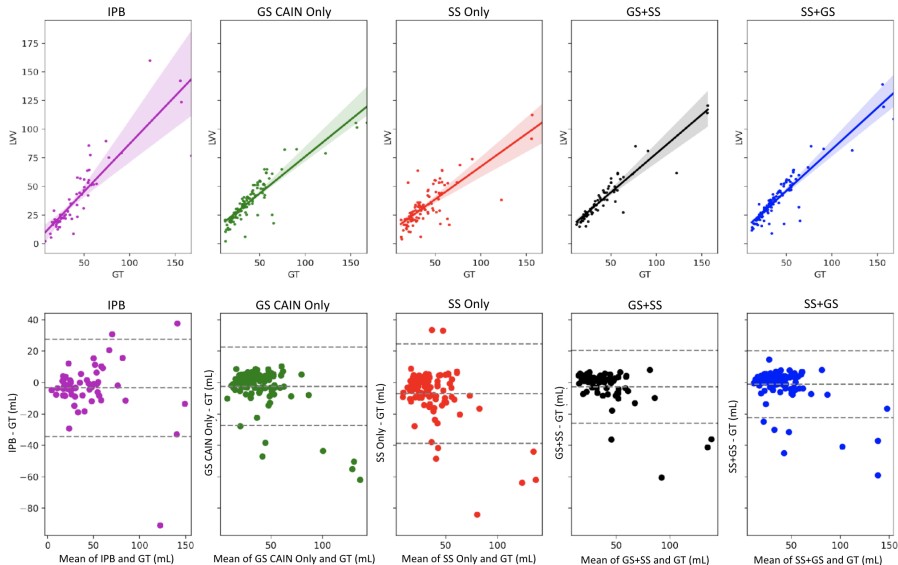

Figure 2: Linear regression and Bland Altman plots between ground truth and predicted LVVs. SS only, GS+SS, and SS+GS are from models with 100 SS masks.

Table 5: Linear regression and correlation coefficients for each model on target datasets.

| Dataset | Model | $R^2$ | Intercept | Slope |
|---------|-------|-------|-----------|-------|
| Source | IPB | 0.78 | -4.07 | 0.93 |
| | GS CAIN Only | 0.95 | **0.94** | **1.02** |
| | SS Only | 0.94 | -3.91 | 0.95 |
| | GS+SS | **0.97** | 3.65 | **1.02** |
| | SS+GS | **0.97** | 1.61 | 1.06 |
| Target | IPB | 0.77 | **6.80** | **0.81** |
| | GS CAIN Only | 0.80 | 12.41 | 0.62 |
| | SS Only | 0.54 | 13.49 | 0.49 |
| | GS+SS | 0.81 | 9.82 | 0.66 |
| | SS+GS | **0.85** | 9.91 | 0.71 |
| ADNI | IPB | 0.84 | **-1.19** | **1.06** |
| | GS CAIN Only | 0.89 | 15.19 | 0.62 |
| | SS Only | 0.68 | 11.92 | 0.48 |
| | GS+SS | 0.86 | 9.33 | 0.64 |
| | SS+GS | **0.95** | 5.13 | 0.81 |
| CCNA | IPB | 0.93 | **4.97** | **0.81** |
| | GS CAIN Only | 0.84 | 12.14 | 0.66 |
| | SS Only | 0.61 | 15.43 | 0.45 |
| | GS+SS | **0.96** | 8.48 | 0.76 |
| | SS+GS | 0.86 | 9.95 | 0.74 |
| ONDRI | IPB | **0.75** | 14.52 | 0.42 |
| | GS CAIN Only | 0.68 | **10.93** | 0.55 |
| | SS Only | 0.57 | 16.22 | 0.42 |
| | GS+SS | 0.45 | 14.52 | **0.58** |
| | SS+GS | 0.71 | 14.31 | 0.55 |

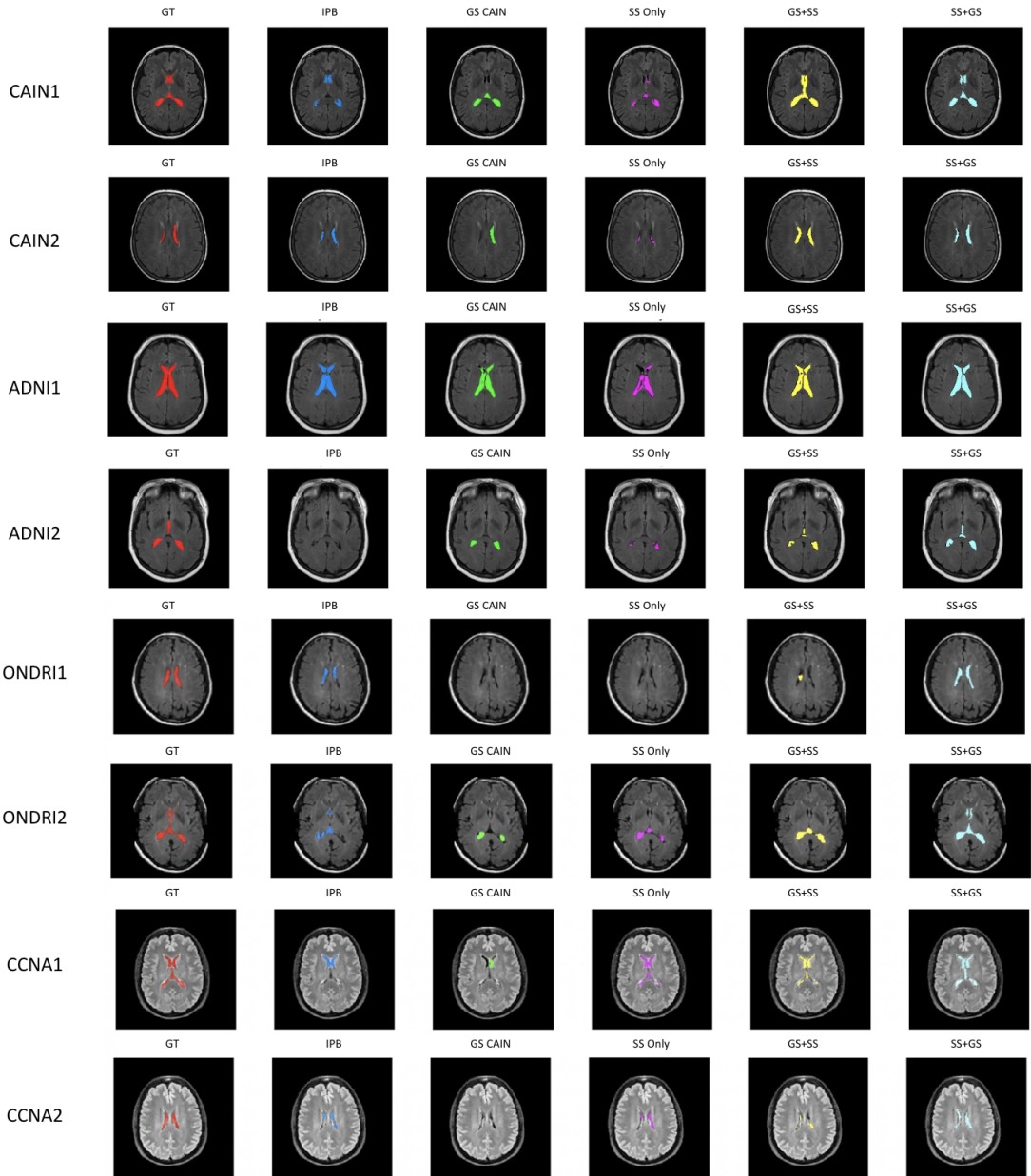

Figure 3: Prediction examples. SS Only, GS+SS, SS+GS models trained on 100 SS masks.

