# OpenReview forum: "Domain Adaptation using Silver Standard Masks for Lateral Ventricle Segmentation in FLAIR MRI"
_MIDL.io/2023/Conference — MIDL 2023 Poster_

### Official Review · Reviewer_QzEg · 2023-02-02

**Confidence:** 4
**Preliminary Rating:** 3

**Summary:**

The authors propose the first Unsupervised Domain Adaptation method for Ventricle Segmentation in FLAIR MRI. This method uses manually annotated ground truth masks from a source domain, and additional automatically-generated masks from an unlabeled target domain to improve the model's performance on the latter. Ventricle volume is an important marker for multiple neurodegenerative diseases.

**Strengths:**


- To the best of my knowledge, the **first Unsupervised Domain Adaptation method for Ventricle Segmentation in FLAIR MRI**. (although contrary to what the authors claim, this is not the first work on LVV segmentation using Deep Learning on FLAIR MRI only see this paper: https://doi.org/10.1016/j.media.2020.101698)
- **Only relies on FLAIR MRI**, which is always **part of the clinical routine**.
- **Thorough analysis** with ablation studies and evaluation of the statistical significance of the results.


**Weaknesses:**


- The **overall clarity** of the paper:
  - Too many acronyms that may confuse the reader (See detailed comments)
  - Section 2.8. Experimental Setup: "4 types of SS masks ×4 datasets (=16)" the different types of SS masks were never defined
  - Some method names: the "all layers" method is actually the method that the authors want to put forward, while the others are just baseline/ablations. This is not made clear by either the name or the writing.
- The **paper is overfull**:
	 - It feels like a few figures that should have been part of the paper have been moved to the appendix to circumvent the 8-page limit. E.g. Fig 6 should have been part of the paper
	 - Even page 4 is overfull and the text overlaps the page number at the bottom of the page
 - **Lack of reproducibility/important technical details**:
   - How was the threshold for generating SS chosen? Does it need to be different for each target domain?
   - When is the fine-tuning of "All layers" models on the src GS data stopped? I.e. how do you prevent overfitting on the src domain?


**Deanonymize Review:**

no

**Detailed Comments:**

The authors' methods itself is not novel (SS masks were already previously used to improve segmentation results https://doi.org/10.1117/12.2581895), but confirm that this is a good solution for leveraging unlabeled data, especially for domain adaptation. The method is also thoroughly evaluated through both ablation studies and statistical tests, which I really enjoyed. However, the paper feels too full and lacks some clarity.

Comments:
- Regarding the title:
	- While definitions of transfer learning (TL) and domain adaptation (DA) may differ from paper to paper, it is commonly accepted that DA is part of TL. So writing "UDA using TL" is redundant and lengthens the title without providing additional information. So I can only suggest changing the title to something along the lines of "Unsupervised Domain Adaptation using Silver Standard Masks for Ventricle Segmentation in FLAIR MRI", which would also explicitly state the type of SS annotation used.

- Regarding the abstract:
   - The authors' method is **not the first Deep-Learning based one for Ventricle Segmentation in FLAIR MRI** (see https://doi.org/10.1016/j.media.2020.101698). It is however the first UDA method for such application.

- Regarding the introduction:
	- There are too many acronyms defined, which increases the mental load for the reader, esp if the acronym is barely used afterward e.g. AD, WML, TBV, UDA
	- The authors claim that they propose a generic "framework". To me, this means that the code for the paper will be published, but I could not find any mention of that. I would suggest changing "framework" to "method".

- Regarding the methodology:
	- **There is critical information missing regarding the threshold**. What's its value? Otherwise, results may not be reproducible. How was it chosen? Was it different for each target domain? If yes, then we still need some (tiny) human input to apply this method to a new domain. Which effect has the choice of threshold on the results? e.g. what happens if we choose too high or too low of a threshold?
	- **There is critical information missing regarding the fine-tuning of "all layers" models**. How was the number of epochs chosen for this step? How do you prevent the models from over-fitting on the src domain?
	- The naming of the 3 types of models could be made more clear from the writing, i.e. that:
		- "GS CAIN Only" is the baseline
		- "SS Only" is the ablation study
		- "All layers" is the authors' actual method. Its name also does not make clear what this method is about, i.e. pre-training+fine-tuning.
	- In experimental setup, when the number of trained models is detailed, 4 types of SS are mentioned. What are these 4 types? It's unclear for the reader and I first assumed that different thresholds were used for generating the SS masks. But it seems the 4 "types" are actually 4 different setups, where the number of SS masks used for training changes.

- Regarding the results:
  - In 3.1. Source Domain Performance: the authors state there was no statistical significance between results from "All layers" models and the GS CAIN Only model.
  - Table 2: the column "Datasets" is not useful, as it remains constant for each model. The ordering of datasets is also irrelevant in this table anyway. The authors then state that "This indicates performance of the model within distribution is maintained or slightly improved", while the statistical test only shows that the performance was only maintained and with no indication that it was slightly improved.
  - Just before Table 3: "Models with 100 SS masks (the best DSC) were retained" it would be clearer to write that "All Layers models with ... were retained"

- Regarding figures:
  -  It feels like the paper is overfull as some figures that should be part of the paper's body have been moved to the abstract e.g. Fig 2, 3. In particular, Fig 6, should be part of the results section and discussed there. Also the heatmap-color choice for fig 6 is unusual, where red is usually used for worse results. I would personally recommend the "Blues" and "crest" color palettes in `seaborn`.
  - **Figure 1 is also never cited**, despite being part of the paper's body.

- Regarding the conclusion:
  - The authors claim that their method is effective for using the vast amounts of unlabeled target data available in imagine centers. However **this contradicts their results**, as there was no statistical difference between the All Layers model trained on 50 SS masks and the model trained using 200 SS masks. So while the authors' method is allows to leverage unlabeled data, their method does allow no leverage a **large** amount of unlabeled data.

- Typos:
  - Bottom of page 1: "(Ertekin et al., 2016)" is cited twice after the last sentence of the 2nd paragraph.
  - Table 1, last column: is the column name "LVV" or "VV"?
  - Inconsistent naming of the U-Net: sometimes it's referred to as "U-Net", sometimes as "UNET"
  - Page 8:
	  -  "multicnetre"
	  - "The method has been successfully applied to multiple target domains with similar results demonstrates the robustness", missing word after "results"
	  -  "This improvement can be seen visually in the examples", missing word
	  -  "The GS CAIN Only model~~s~~ underestimate**s** and miss**es** small ventricular areas"


**Paper Type:**

validation/application paper

**Questions To Address In The Rebuttal:**


- Regarding the choice of threshold for generating SS masks:
  - What is the chosen threshold value?
  - How was it chosen?
  - Was it different for each target domain?
  - Which effect has the choice of threshold on the results? e.g. what happens if we choose too high or too low of a threshold?
 - Regarding fine-tuning "all layers" models:
	- How was the number of epochs chosen for this step?
	- Is there any metric or criterion for early stopping?
	- How do you prevent the models from over-fitting on the src domain?
- Will the code for the paper be made publicly available? (see framework claim)


The double-blinded review rule is broken, but it's fine according to the guidelines. So, I am takin a step-up in my recommendation.

---

### Official Review · Reviewer_vVtN · 2023-02-06

**Confidence:** 3
**Preliminary Rating:** 3
**Recommendation:** Poster

**Summary:**

This paper proposed an unsupervised domain adaptation method for the segmentation of lateral ventricular volume of FLAIR images. The proposed unsupervised domain adaptation method improved the accuracy and consistency of the segmentation compared to the prediction from the source domain only model. The idea is generic, and can be applied to any other segmentation tasks with multi-site dataset.

**Strengths:**

This paper is overall well written and easy to follow. The authors demonstrated the improvement of the proposed method through extensive experimental results. The proposed method is clinically significant since the dataset is usually acquired from multi-center/site and different scanners.

**Weaknesses:**

It seems that the generation of silver standard masks is a critical step in the proposed method. It needs more details such as what value was set for a threshold, and how it was determined. What is the kernel size for erosion? What kind of kernel was used? What kind of distance was used for the computation of mid-sagittal plane? How was anterior-posterior commissure line decided?

**Deanonymize Review:**

no

**Detailed Comments:**

-In section 2.3, is Gold standard mask only generated for source domain? How was GS annotation done for the target domain? Please clarify.

-Please move Table 3 to the top of the paper. Currently it is located in the middle of the paragraph, which looks weird.


**Paper Type:**

methodological development

**Questions To Address In The Rebuttal:**

-Using silver standard masks in the target domain means that the authors have partial information about the target, meaning that it’s not completely unsupervised. Unsupervised learning usually refers to the task without any labels such as clustering or auto-encoders. I think this method can be considered as weakly-supervised.
-Generation of gold standard masks are laborious and time-consuming. How about silver standard masks? How long does it take to get silver standard masks?
-What is the input image resolution for U-Net?
-Can the authors provide the quantitative results for SS masks for the target test set without any training of U-Net, since SS mask was served as an initial guess?

---

### Official Review · Reviewer_DQZk · 2023-02-06

**Confidence:** 4
**Preliminary Rating:** 5
**Recommendation:** Best Paper Award, Oral, Poster

**Summary:**

This paper describes a transfer learning method between a source domain in which gold standard labels (manually annotated by radiologists) are available, and target domains in which only silver standard labels (computed by an unsupervised method) are available. The authors developed a new unsupervised method to produce the silver standard labels for their application, which is the segmentation of the lateral ventricles.
The method is extensively validated using 4 research data sets of FLAIR MRI of (slightly) demented patients. The validity of the targeted measurement extracted from the segmentation, the lateral ventricular volume, was also assessed.

**Strengths:**

- The methods are well described, and the validation procedure is sound.
- The developed method was compared to variations without gold or silver labels while keeping the same amount of volumes, thus allowing a fair comparison. The performance of the unsupervised annotation method is also given, and is indeed lower than the deep learning ones.
- The figures and tables are helpful to understand the content.

**Weaknesses:**

- the gold labels were done by a medical student trained by a radiologist. It could be worth to know if there is a high inter-rater difference between these and the annotations of a trained radiologist.
- the method developed by the authors consist in first pre-training on the target data and then fine-tune on the source data. It could be worth knowing if the same improvement is noticed by inverting the two steps (pre-training on source and fine-tuning on target) as it is what is usually done.

**Deanonymize Review:**

yes

**Detailed Comments:**

- the unsupervised method relies on the fact that the ventricles should be located in the middle of the brain. Though this is usually the case, it may not work for patients with a significant midline shift (due for example to a tumor or a hematoma). This comment could be added in the discussion.
- The source data set CAIN is the one with the smallest mean ventricular volume. This discrepancy, especially with ADNI, could be discussed. Would have the things been different if another source data set was chosen?
- There are still some typos (for example there is no verb in the first sentence of the abstract).

**Paper Type:**

both

**Questions To Address In The Rebuttal:**

- Additional experiments inverting the two steps of the transfer learning procedure (first source then target) would further improve the paper.
- Though there is no automatic method for ventricular segmentation on FLAIR MRI, it would be beneficial to add a value obtained by automatic methods on other modalities (T1-MRI?) to get an idea what was already achieved for a similar task.

---

### Meta-Review · Area_Chair_MaLa · 2023-02-21

**Recommendation:** Accept (Poster)
**Confidence:** 4

**Metareview:**

Interesting idea but not necessarily novel work for domain adaptation that is based on pretraining with a silver standard in the target domain obtained using unsupervised learning and finetuning with a limited number of high-quality segmentation masks in the source domain. The reviews are generally positive, with one reviewer even recommending the paper as a strong accept. Authors have extensively replied to the reviewer comments and even added new results. The text added to the abstract is not entirely self-contained (what is CAIN?). The paper contains some typos and grammatical errors, a quick run-through with a tool like Grammarly would likely resolve most of those before a camera-ready version.

There is a bit of a semantic discussion with one of the reviewers about using the word ‘unsupervised.’ I concur with the reviewer that it would indeed be better not to call the method used to obtain the silver standard ‘unsupervised.’ The silver standard is obtained using a ‘conventional’ image processing algorithm, and I think the term ‘silver standard’ is adequate, but ‘unsupervised’ implies that some kind of learning is used to obtain these (which is not the case).